# Molecular Mechanisms of Glucocorticoid-Induced Insulin Resistance

**DOI:** 10.3390/ijms22020623

**Published:** 2021-01-09

**Authors:** Carine Beaupere, Alexandrine Liboz, Bruno Fève, Bertrand Blondeau, Ghislaine Guillemain

**Affiliations:** 1INSERM UMR_S938, Saint-Antoine Research Center, Hospitalo-Universitary Institute, Sorbonne Université, ICAN, 75012 Paris, France; carine.beaupere@inserm.fr (C.B.); alexandrine.liboz@gmail.com (A.L.); bruno.feve@inserm.fr (B.F.); 2Assitance Publique-Hôpitaux de Paris, Service d’Endocrinologie, Hôpital Saint-Antoine, 75012 Paris, France

**Keywords:** insulin resistance, glucocorticoids, signaling pathway, liver, muscle, adipose tissue, pancreatic beta cells

## Abstract

Glucocorticoids (GCs) are steroids secreted by the adrenal cortex under the hypothalamic-pituitary-adrenal axis control, one of the major neuro-endocrine systems of the organism. These hormones are involved in tissue repair, immune stability, and metabolic processes, such as the regulation of carbohydrate, lipid, and protein metabolism. Globally, GCs are presented as ‘flight and fight’ hormones and, in that purpose, they are catabolic hormones required to mobilize storage to provide energy for the organism. If acute GC secretion allows fast metabolic adaptations to respond to danger, stress, or metabolic imbalance, long-term GC exposure arising from treatment or Cushing’s syndrome, progressively leads to insulin resistance and, in fine, cardiometabolic disorders. In this review, we briefly summarize the pharmacological actions of GC and metabolic dysregulations observed in patients exposed to an excess of GCs. Next, we describe in detail the molecular mechanisms underlying GC-induced insulin resistance in adipose tissue, liver, muscle, and to a lesser extent in gut, bone, and brain, mainly identified by numerous studies performed in animal models. Finally, we present the paradoxical effects of GCs on beta cell mass and insulin secretion by the pancreas with a specific focus on the direct and indirect (through insulin-sensitive organs) effects of GCs. Overall, a better knowledge of the specific action of GCs on several organs and their molecular targets may help foster the understanding of GCs’ side effects and design new drugs that possess therapeutic benefits without metabolic adverse effects.

## 1. Introduction

Type 2 diabetes mellitus (T2DM) is a multifactorial disease that associates insulin resistance and decreased insulin secretion. Among factors that control insulin sensitivity and insulin secretion, glucocorticoids (GCs) have been implicated in studies in Human and the molecular mechanisms involved have been deciphered in animal and cell culture studies. Here, we present a review of the effects of GCs on insulin sensitivity and secretion through a staggered analysis of GC action on the main organs responding to insulin, but also on the pancreas. Clear evidences point to GCs as potent activators of insulin resistance and inhibitors of insulin secretion, but recent evidences also identified inter-organ communications, at least in mice, linking GC-induced insulin resistance with improved insulin secretion.

## 2. Glucocorticoid Production and Action

GCs are steroids naturally secreted by the adrenal cortex from cholesterol under the hypothalamic-pituitary-adrenal (HPA) axis control, one of the major neuro-endocrine systems in the organism. The secretion of cortisol follows a circadian rhythm and is involved in tissue repair, immune stability and metabolic processes. Additionally, under psychological and physiological stresses, the HPA axis triggers the neuro-endocrine stress response resulting in a massive adrenal release of cortisol and the regulation of carbohydrate, lipid and protein metabolism. High concentrations of GCs also induce a negative feedback loop on the HPA axis to stop the response and maintain the homeostasis of the system. Defective interactions between stress response and cortisol circadian rhythm are known to induce numerous deleterious effects on health ranging from depression to obesity and metabolic syndrome [1]. Eighty to ninety percent of estimated cortisol circulate in the organism bound to the corticosteroid-binding-globulin (CBG or transcortine), a glycosylated plasma protein responsible for the transport of steroids, 5% of cortisol remains free in the circulation while the remaining cortisol is bound to albumin [2]. CBG-bound cortisol is attracted by pro-inflammatory chemokines and delivered at the local site by cleavage of the CBG-cortisol binding thanks to the elastase secreted by neutrophils [3,4]. In addition to the regulated production of cortisol by the HPA axis, the concentration of active GC can be modulated locally by two enzymes: type 1 and type 2 11-beta-hydroxysteroid-dehydrogenases (11b-HSD1 and 2, respectively) [5]. In humans, 11b-HSD1 converts inactive cortisone (11-deoxycorticosterone in rodents) to the active cortisol (corticosterone in rodents) capable of binding and activating its receptors. Alternatively, 11b-HSD2 inactivates cortisol into cortisone (or corticosterone to 11 deoxycorticosterone in rodents). *HSD11B1* is expressed in most cells and tissues, with high levels of expression to be found in liver, muscle, and adipose tissues, suggesting that cortisol biodisponibility could be increased in those tissues. *HSD11B2* is principally expressed in kidney, colon, and placenta and reduces cortisol biodisponibility. Since they are lipophilic, GCs can cross the cellular membrane in order to bind their intracellular receptors, the glucocorticoid receptor (GR) and the mineralocorticoid receptor (MR). The affinity of MR for cortisol is higher than the affinity of GR, mediating most of its low dose, physiological effects. However, MR is not highly expressed in all cells and tissues and contrary to the GR, binds different types of steroids, including aldosterone and progesterone. Therefore, in response to an acute GC secretion, after the MR reaches near saturation, most of the stress response is induced by the binding of GCs to the ubiquitous GR [6]. In humans, the GR is encoded by the *NR3C1* gene containing nine exons, from which exons 2–9 encode the GR. The mechanisms of regulation of *NR3C1* expression and translation are complex and tightly regulated in tissues. Up to 13 different promoter regions have been identified in the untranslated exon 1 of *NR3C1* [7], affecting the transcription, the translation and the stability of mRNAs. Interestingly, the promoter regions are differentially expressed within tissues, suggesting tissue-specific regulations of *NR3C1* expression and translation into proteins [8,9]. Additionally, different splice-variants of *NR3C1* transcripts exist, modulating GR function and affinity to GCs. GRα which results from the alternative splicing of exon 8 to exon 9a and GRβ from the alternative splicing of exon 8 to exon 9b in Human, are the most extensively studied isoforms of the GR [10]. GC are known to bind to GRα but not to GRβ, the latter being constitutively nuclear [10,11]. GRα and GRβ are known to induce or repress different sets of genes, GRβ is also known to compete with and inhibit GRα action, participating in GC resistance [12]. Eight different translation initiation start sites exist in the exon 2 of the human *NR3C1* gene, consequently eight different subtypes of GRα can be produced named GRαA, B, C1-3, and D1-3. All of the GRα subtypes have specific expression patterns. As observed in rat and mice, GRα-C isoform is more expressed in pancreas and lung compared to liver, GRα-D is preferentially expressed in spleen and bladder, whereas GRα-B isoform is more expressed than the GRα-A isoform in the liver and thymus, suggesting specific functions of the GRα subtypes in tissues or cells [13]. Those variations are especially relevant considering the observation from the same authors that the different GRα subunits regulate different sets of genes upon binding to GCs [14]. Additionally, GR proteins undergo different post-translational modifications, such as phosphorylation, acetylation, methylation, ubiquitination, and SUMOylation modifying its functionality [15]. The ligand-inducible GRα is located in the cytosol as a molecular complex with heat shock proteins HSP90, HSP70, HSP56, P23, and immunophilin FKBP51. The binding of GC to GRα exposes the GR nuclear localization signal and allows the translocation of the GC-activated transcription factor into the nucleus. The genomic effect of GR in the nucleus can either be direct via the binding of GR on specific positive or negative binding sites in DNA, named GRE (glucocorticoid response element) or negative-GRE (nGRE) or indirectly via the interaction of GR with other proteins or transcription factors, by a mechanism known as tethering [16]. Due to the presence of nGRE and GRE within the promoter and coding sequence of the *NR3C1* gene, activated GR can modulate its own expression, participating in a negative feedback loop when binding to GCs. Activated GR regulates the expression or activity of many genes and transcription factors involved mainly in metabolic and immune processes and including *NF-κB*, *AP-1*, *STATs*, *C/EBP*, and *PPAR* [16].

## 3. Pharmacological Use of Glucocorticoids and Main Side Effects

GCs display powerful anti-inflammatory and immunosuppressive capacities. Therefore, synthetic GCs are extensively used for the treatment of numerous autoimmune and inflammatory diseases, in the fields of pneumology, dermatology, endocrinology, rheumatology, gastroenterology, or oncology. Several forms of synthetic GCs have been developed and allow a wide range of options for intensity, duration, and MR over GR potency. For example, prednisolone, the most commonly used GC, displays a mineralocorticoid over glucocorticoid activity ratio (relative to cortisol) superior to the synthetic glucocorticoid dexamethasone but a shorter duration of activity, making it suitable for a short-term treatment of patients without fluid or electrolyte abnormalities (e.g., water and salt retention). In contrast, the powerful mineralocorticoid over weak GC potency of fludrocortisone is used to treat Addison’s disease and salt waste pathologies as a substitute to the lacking aldosterone to induce water and sodium retention, as well as blood pressure normalization [17]. Along with the improved quality of life provided by their therapeutic use, GC exhibit numerous side effects on several physiological, metabolic, and even psychological parameters (depression, irritability, insomnia, etc.) and remain among the most misused drugs. Adverse effects of GCs sorely depend on the dose, the type of steroids used, and the duration of treatment, although the preexisting conditions of the patient influence the outcome, suggesting the existence of vulnerable populations. In this review we will focus on the metabolic effects of GC upon adipose tissues, liver, muscle, pancreas, and other tissues.

GC therapy is frequently associated with hyperglycemia, fat deposits redistribution and insulin resistance (Box 1). Thus, they are the main drugs responsible for the so-called steroid-induced diabetes. However, the routes of administration of GC do not have the same outcomes on diabetes incidence, with generally a lower association for long-term inhalation, topical application, or intra-articular injections. Yet, up to 2% of diabetes cases are associated with oral or intravenous GC therapies [18]. An investigation on the prevalence of risk factors for hyperglycemia in hospitalized patients receiving GCs shows that the majority presents hyperglycemic episodes, with more occurrences in patients with a diabetes history [19]. Patients undergoing chronic or prolonged oral/intravenous cortico-therapy should be closely monitored as it may lead to hyperglycemia over time [20]. It has been shown that if the development of insulin resistance may take up to one or two days of treatment, a single acute exposure to GCs induces glucose tolerance alterations within 12h in healthy volunteers [21]. The origins of these glucose metabolism abnormalities are complex and involve multiple organs.

Apart from their metabolic actions, GCs act on bone formation and GC treatments are the secondary most common cause of osteoporosis. Yet, skeletal adverse effects of long-term GC therapies remain underestimated. Of concern are high doses and long-term GC treatments in children and adults displaying other skeletal risk factors (example: post-menopausal women) for whom bone mass loss and fragility can be more deleterious. Interestingly, there is a higher incidence of vertebral fractures among newly GC-treated patients (<6 months) but a relative decline in vertebral fractures after longer exposure to GC (>6 months) [22]. Moreover, based on a database of newly GC-treated rheumatoid arthritis patients aged 18-64 years (mean age 49 years), Balasubramanian et al. demonstrated a higher fracture risk among younger, newly-treated patients and showed a 2–2.5-fold increased fracture risk for patients receiving higher doses of GCs compared to low doses receivers. Whether these effects on bone are involved in glucose homeostasis or insulin sensitivity dysregulations induced by GCs is unknown in humans, but will be presented in the paragraph on animal models of GC treatment.

In humans, it is well known that a long-term GC treatment induces weight gain and adipose tissue redistribution in connection with major changes in glucose and lipid metabolism. The development of these Cushingoid features (abdominal obesity, dyslipidemia, cervical fat deposits, insulin resistance, hyperglycemia, etc.) depends on the dose and duration of treatment. In a study where 2167 subjects undergoing more than 60 days of a mean 15 mg/day of prednisolone, up to 80% report weight gain and almost 10% exhibit hyperglycemia at a higher dose of prednisolone [23]. In fact, GCs are reported to have both lipolytic and adipogenic actions driving lipodystrophies in patients with an expansion of the visceral adipose tissue (VAT) and a reduction in the subcutaneous one (SCAT), greatly increasing the cardiometabolic risk in patients [23,24]. Moreover, GC favor free fatty acid release from adipose tissues in the circulation, causing ectopic storage of fat in other organs, such as liver and skeletal muscle, participating in those tissue dysfunctions [23,24].

Muscles contain the main glycogen reserve of the body and absorb up to 80% of postprandial glucose under insulin control. Thus, GC-induced impairment of skeletal muscle function contributes to glucose homeostasis disturbance. Another feature of short- or long-term GC-therapy is the reduction of the muscle mass by enhanced protein degradation and an elevated amino acids circulation [24]. Thereby, global loss of muscle tissue capable of glucose uptake and massive release of amino acids contribute to a vicious circle of maintenance of insulin resistance and hyperglycemia.

In the liver, GCs enhance endogenous glucose production by activating numerous genes involved in carbohydrate metabolism, like *phosphoenolpyruvate carboxykinase* (*PEPCK*) and *glucose-6-phosphatase* (*G6Pase*), inducing gluconeogenesis [25]. The lipolysis and muscle protein degradation induced by long-term GC exposure also provide substrates for gluconeogenesis. Furthermore, direct interference with insulin signaling and elevated triacylglycerol (TAG) supply to the liver are responsible for hepatic insulin resistance [26].

Insulin secretion in humans may be severely affected by GCs despite a great adaptation capacity of beta cells [27]. Although only short pharmacologic exposure (up to 14 days) to GC can be assessed in Human due to ethical limitations, some interesting studies highlight the impact of exogenous GC and/or chronic endogenous hypercorticism on pancreatic endocrine functions. Among patients suffering from Cushing’s syndrome, over 30% will develop diabetes [28], associated with an impaired glucose tolerance and a decrease in insulin sensitivity prior to the clinical stage of the disease, suggesting that some individuals are more likely prone to develop diabetes in response to supraphysiological GC doses. In fact, a hyperglycemic clamp study performed on healthy subjects treated with the synthetic GC dexamethasone (DEX) (15 mg/d for two days) reveals that high-insulin responders (HIR; with a strong insulin response to a glucose load) show increased insulin release compared to low-insulin responders (LIR). Moreover, DEX treatment increases insulin response in HIR but not in LIR subjects despite similar DEX-induced decrease in insulin sensitivity in both groups. Similarly, LIR subjects display higher fasting blood glucose and blood glucose increase in response to OGTT, with 50% of LIR subjects presenting an impaired glucose tolerance after two days of DEX treatment [29]. This princeps study shows that individuals with an already deficient insulin secretion capacity are more susceptible to develop steroid-induced diabetes [27]. In Cushing’s syndrome or following a two-week exposure to GC, subjects show high insulin plasma levels at a basal state, but fail to increase their insulin response to a glucose load or standardized meals [30,31,32]. This phenomenon, called “relative hypoinsulinemia”, is an indicator of the direct or indirect noxious effects of GC on beta cells. Moreover, GCs modulate glucagon secretion as another counter-action with insulin activity. In a study on healthy non-obese subjects, DEX treatment (2 mg/d for three days) results in a 55% increase in basal glucagon levels and in a 60-100% increase in the maximal glucagon response to alanine infusion or protein ingestion [33]. Moreover, patients with Cushing’s syndrome show, respectively, 100% and 170% increase for the same parameters without DEX treatment, suggesting a persisting hyperglucagonemia in chronic hypercorticism [33]. Thus, gluconeogenic and diabetogenic effects of GCs are also directly mediated by their action on the pancreatic endocrine function, with decreased beta cell response in susceptible subjects and increased alpha cell activity in all subjects.

Box 1Clinical Metabolic Consequences of GC Overexposure.To summarize, GCs induce hyperglycemia and glucose intolerance and are responsible for 2% of diabetes, called steroid-induced diabetes. Long-term GC-induced metabolic adverse effects include weight gain; deleterious fat redistribution, and increased free fatty acid circulation; reduction of muscle mass and massive release of amino acids; enhanced gluconeogenesis and endogenous glucose production; bone mass loss and higher fracture risk; most of these features are caused directly by the negative impact of GCs on pancreatic endocrine functions and peripherical insulin sensitivity.

## 4. Direct and Indirect Effects of Glucocorticoids upon Pancreatic Endocrine Function

GC are potent regulators of metabolism and have thus been extensively studied in regards to their effect on insulin secretion by pancreatic beta cells. Yet, the situation is still unclear and there is, for example, still a controversy about the expression of both 11b-HSD enzymes in pancreatic beta cells while a recent study showed that alpha cells lack 11b-HSD [34,35]. Moreover, there is a paradox between in vitro and in vivo models regarding the response of pancreatic beta cells to GCs and we will focus on describing how this paradox may be solved (Box 2).

In vitro, the majority of studies show a negative effect of GCs on the secretion, proliferation, as well as on the survival of beta cells. Meanwhile, the signaling pathway altered in beta cells are still not well known [36] (Figure 1). A study on isolated human and murine islets as well as on Min6 cells (a mouse pancreatic beta cell line) shows that addition of DEX or prednisolone leads to a decrease in cell viability via activation of the p38 MAPK/TXNIP pathway and that this effect was reversed by the blockade of the GR [37]. More studies on DEX-induced beta cell apoptosis show several pathways involved in response to GR activation in beta cells [38,39,40,41]. In vitro, beta cell insulin secretion is severely impacted by GC exposure. DEX impairs beta cell response to glucose by decreasing GLUT2 protein level, by a post-translational mechanism [42], but also the insulin release: the upregulation of transcription and expression of *SGK1* increases repolarizing activity of K^+^ channels, reducing then Ca^2+^ channels activity and insulin vesicles exocytosis [43]. Accordingly, INS1 (a rat pancreatic beta cell line) treatment with prednisolone results in an induction of endoplasmic reticulum stress and impaired insulin biosynthesis and release. Prolonged exposure to GC induces the unfolded protein response (UPR) and, *in fine*, beta cell apoptosis [44]. However, a recent study further described the negative impact in vitro of GC treatment. Fine et al. showed in human and murine isolated islets treated with GCs that the suppression of the calcium channel activity did not impaired insulin secretion or beta cell response to glucose. In fact, they identified an enzymatically amplified feedback loop whereby GCs boost cAMP to maintain insulin secretion in the face of perturbed ionic signals [35]. They also emphasized that lipotoxicity prevented this cAMP increase by GCs, thus suppressing insulin secretion improvement. Knowing that systemic supraphysiologic GC levels lead to lipolysis and dyslipidemia, it is important to consider integrated systems to better apprehend influence of GCs on beta cell function.

In vivo models show a wide heterogeneity in the pancreatic response to GC exposure associated with a complex metabolic board involving multiple organs. They allow functional and morphological studies of pancreas and beta cells upon GC treatment to clearly define the adaptation of beta cell mass and function. One study using transgenic mice overexpressing GR under the control of the insulin promoter shows a decrease in glucose tolerance due to impaired insulin release, demonstrating that GC signaling has a direct inhibitory effect on glucose-induced insulin release in vivo [45]. This result could be explained by the need for a whole-organism adaptation induced by stress and requiring high blood glucose to provide sufficient energy to muscles and the brain to support the fight-or-flight mechanism. However, if acute exposure to low doses of GCs is not detrimental for the beta cell mass and function, longer exposure or higher doses will trigger adaptive mechanisms to preserve euglycemia. Numerous studies in rodents acknowledge beta cell adaptation through proliferation in response to multiple states of insulin resistance like pregnancy, high-fat diet, partial pancreatectomy or exposure to GCs. However, the pathways involved are still debated. Rats treated for one day with DEX show pancreatic adaptation before the onset of marked insulin resistance with hyperinsulinemia and augmented glucose-stimulated insulin secretion (GSIS). After 3–5 days of treatment, markers of insulin resistance are associated with hyperinsulinemia and beta cell hyperplasia. This adaptation of beta cell mass is time- and dose-dependent, as only an increase in insulin secretion is necessary to counteract short-term or low-dose GC treatment effects [46,47]. The combination of corticosterone exposure and high-fat diet (HFD) in young rats first increases insulin secretion and beta cell mass by proliferation. However, HFD and corticosterone act synergistically to promote severe insulin resistance beyond the adaptive capacity of beta cells, resulting in impaired insulin response to glucose and hyperglycemia in a longer run [48]. Our team recently proposed that beta cell mass may adapt to GC-induced insulin resistance through beta cell neogenesis, that is the formation of new beta cells from precursors. C57bl6/J adult mice treated for four weeks with corticosterone exhibit severe insulin resistance associated with expansion of the beta cell mass both by proliferation and neogenesis, as well as increased insulin secretion at a basal state and in response to glucose stimulation [49]. It also seems that enhanced GC sensitivity, induced by the overexpression of the GR in beta cells, can worsen aged-induced insulin resistance and promote the development of diabetes in one-year-old mice [50]. Moreover, these mice present impaired glucose tolerance at three months, progressing to a diabetic state at 12 months of age, suggesting that GC exert a strong and direct diabetogenic effect on beta cells, potentially through the regulation of insulin secretion via the α2-adrenergic receptor [50].

While GR overexpression in mature beta cells does not alter proliferation but impairs insulin secretion, GR overexpression in precursor cells decreases adult beta cell fraction but did not change insulin secretion or glucose tolerance [51]. In fact, GCs impact pancreatic beta cell formation and function early in embryonic development. Deletion of *NR3C1* in pancreatic precursor cells with a Cre-recombinase under Pancreatic and duodenal homeobox 1 (*PDX1*) promoter led to a doubled beta cell mass with increased islet number and size, defining GC as major inhibitors of beta cell development [52]. Furthermore, we also defined PGC-1alpha, a co-regulator of the GR, as a main actor of GC actions on insulin secretion [53], with a specific impact on mitochondrial function and oxidative stress [54]. Yet, windows of exposure of beta cells to GCs may also define their actions on this cell type. For at least four decades, a low birth weight has been known to be associated with higher risk of metabolic diseases such as diabetes in adulthood [55,56]. We have previously provided evidence linking maternal and fetal undernutrition with increased maternal and fetal corticosterone levels, resulting in a low birth weight and a decreased beta cell mass associated with glucose intolerance in adulthood [57,58]. These studies demonstrated the importance of GC upon fetal development of the beta cell mass from precursors during specific time-frame. Additionally, *NR3C1* deletion in these precursors reversed the negative impact of the fetal undernutrition/corticosterone overexposure on the beta cell fraction in adulthood. More recently, we provided new insights on the role of fetal GCs on beta cell function in humans: we characterized the insulin sensitivity and secretion in young adults that were exposed to high GC doses during their early fetal life. In fact, such treatment is used to prevent androgen overproduction by the adrenals in cases of congenital adrenal hyperplasia due to 21-hydroxylase deficiency. Yet, such treatment also involved that fetuses that did not carry the mutation leading to the disease were still exposed in utero to high GC doses. We explored such exposed adults and compared them to age- and BMI-adjusted controls. We found that young adults who were exposed to high GC doses presented with a normal insulin sensitivity but a reduced insulin secretion in response to glucose. Thus, the results that were obtained in rodents were confirmed in humans and define GCs as potent effectors of the fetal programming of beta cell dysfunction [59].

Box 2Paradoxical Effects of GCs on Beta Cell Function and Mass In Vitro and In Vivo.In vitro, GCs decrease beta cell viability, reduce insulin secretion, and release and impair insulin synthesis. In vivo, complex modifications implicating inter-organ communications generate a wide heterogeneity in the response of endocrine pancreas to GC depending on the model, dose, and duration of GC treatment.GC treatment on adult animal models may inhibit glucose-induced insulin release (Figure 1) but other studies demonstrated adaptive mechanisms to maintain/increase beta cell mass by proliferation or neogenesis resulting in improved insulin secretion and glucose tolerance. Interestingly, fetal exposure to GCs decreases adult beta cell fraction and ultimately leads to impaired insulin secretion later in adult rodents and humans.

Thus, GCs control glucose homeostasis, and dysregulation of GC levels are associated with metabolic changes and blood glucose alterations. Moreover, GCs interfere with insulin signaling and it has been proved that GCs are able to modulate insulin signaling at the molecular level.

## 5. A Brief Molecular View on the Insulin Signaling Pathway

The objective of the present review is to precisely describe the molecular mechanisms of GC-induced insulin resistance. Before, a brief description of the insulin signaling is required. Under normal physiological conditions, after consuming a meal, glucose and amino acids are transported from the intestine to the bloodstream. This rise in blood glucose stimulates pancreatic beta cells to secrete insulin. The hormone binds to its receptor located on membranes of target tissues. Upon insulin binding, its receptor autophosphorylates and phosphorylates insulin receptor substrates (IRS), further leading to activation of the phosphoinositide-3-kinase (PI3K), resulting in the phosphorylation of protein kinase B or AKT that finally activates pathways regulating insulin actions in target tissues. Insulin resistance occurs when a higher dose of insulin is required to produce the same effects on glucose storage. Consequently, the glucose uptake and utilization by liver, muscles and adipose tissues is decreased, while the hepatic glucose production is increased, resulting in hyperglycemia and, eventually, type 2 diabetes. We will now present molecular events that underlie insulin resistance induced by GC in the main organs involved in glucose homeostasis and insulin action, with a specific focus on the molecular mechanisms identified in animal models and in vitro studies.

## 6. Long-Term Glucocorticoid Exposure and Insulin Resistance of The Adipose Tissue

### 6.1. Glucocorticoids and Insulin Resistance in Adipose Tissues In Vivo

Adipose tissue is an organ composed of different depots found in several locations throughout the body and differing in developmental, structural and functional features. The four main functions of the adipose tissue are energy storage in the form of TAG, release of free fatty acids (FFA) and glycerol, and the secretion of adipokines and thermoregulation. The function of some particular fat depots is governed by their differential cellular composition, including adipocytes but also a stromal vascular fraction composed of preadipocytes, mesenchymal progenitors, endothelial and immune cells [60]. Traditionally, three types of adipocytes—white, brown and beige—are found in the adipose tissue. White adipocytes are classical adipocytes specialized in the storage and release of fatty acids. The brown fat, found mainly in the interscapular and perirenal regions in mice and in the supraclavicular, paravertebral, and perirenal regions in humans is involved in the thermoregulation and dissipate energy in the form of heat. Brown adipocytes are rich in mitochondria and in uncoupling protein-1 (UCP-1) activity responsible for the thermogenesis. In rodent, prolonged exposure to cold provoke the appearance of *UCP-1*-expressing cells within white fat depots, these inducible cells with brown-like features are the beige adipocytes. Without cold or beta-adrenergic stimuli, beige cells function more as white adipocytes and are involved in energy storage rather than dissipation. White adipose tissues (WAT) represents the majority of the adipose mass in rodents and comprise the visceral fat (perirenal, peritoneal, mesenteric, gonadal) and subcutaneous fat (inguinal, subscapular) [61]. In addition to those major WAT depots, more discrete white fat depots can be found near or inside various tissues such as bone marrow, mammary or hypoderma, allowing different metabolic regulations locally [60,62,63]. Interestingly, in rodents, subcutaneous fat transplanted in the visceral area improves glucose metabolism and insulin sensitivity indicating the implication of adipocyte cell-specificities rather than anatomical differences [64]. The adipose tissue is involved in glucose homeostasis through different functions: (1) lipogenesis, i.e., *de novo* fatty acid synthesis from glucose, allowing fat storage as TAG; (2) glucose and fatty acid oxidation within brown and beige adipocytes for thermogenesis; and (3) via the secretion of endocrine factors influencing the function of other tissues involved in the glucose homeostasis such as liver, muscle, and pancreatic beta cells. Indeed, the disturbed fat storage into WAT observed in lipodystrophies, obesity, or insulin resistance leads to the ectopic accumulation of fat in other tissues such as skeletal muscle, liver, pancreas, driving the insulin resistance or dysfunction in these tissues [65]. In addition to white adipocytes, brown and beige adipocytes are sensitive to insulin. As such, constitutive insulin receptor (*IR*) deletion in adipocytes leads to lipodystrophy, liver steatosis, and insulin resistance [66]. In contrast, *IR* deletion in brown adipocytes leads to a decrease in insulin secretion and a progressive glucose intolerance but without insulin resistance [67]. In Human, long-term exposure to GC causes a lipodystrophy characterized by an increase in VAT and a decrease of the SCAT [68]. It has been shown that GR is more expressed in visceral adipose tissue (VAT) than in sub-cutaneous adipose tissue (SCAT) [69], and that its activation generates specific effects depending on the action on targeted tissues in humans. In rodents, a long-term treatment with corticosterone in rat and mice induces an increase in visceral fat, an increase in adipogenesis, through preadipocyte differentiation rather than adipocyte hypertrophy and promotes an increase in lipolysis, FFA and glycerol concentrations in the blood [70,71,72,73]. Moreover, GCs are known to reduce the expression and the activity of the cytosolic isoform of PEPCK (PEPCK-C) in epididymal fat depots, thus reducing glyceroneogenesis, which is required for the generation of glycerol-3-phosphate and the formation and the storage of TAG. In contrast, the same DEX treatment in normally-fed rats failed to reduce the PEPCK-C activity in retroperitoneal (white) and interscapular (brown) adipose tissues, indicating fat depot-dependent differential responses to GCs [74]. The molecular mechanisms governing GC’s impact on the different fat tissues are being investigated, but are still poorly understood. Mainly, the local concentration in corticosterone, as regulated by the secretion of GCs by the HPA, but also by the local expression of 11b-HSD enzymes and the downstream signalization trough the glucocorticoid receptors GR and MR, are major determinants of the cellular and tissue response to GC (Figure 2). 11b-HSD1, that converts cortisone into its active form cortisol in Human and 11-dehydrocorticosterone to corticosterone in rodent, is expressed in the adipose tissue, indicating that locally the concentration of active GCs might be higher compared to the circulating one [75,76]. Moreover, GCs and insulin induce 11b-HSD1 expression in the white epididymal adipose tissue amplifying the detrimental impact of long-term exposure to GCs [77,78,79] (Figure 2). Accordingly, adipose-specific *HSD11B1* KO mice are protected from hepatic steatosis and show a decrease in circulating FFA, demonstrating a protection from GC-induced insulin resistance [80,81]. Locally, activated GC can bind to either MR or GR. MR expression and aldosterone levels (locally and in the plasma) are elevated in the visceral adipose tissue of obese patients [82]. Moreover, upregulation of the MR in adipocytes in vivo led to an increase in fat mass and insulin resistance in a mouse model of diet-induced obesity [82]. Pharmacological MR blockade prevented VAT senescence, mitochondrial dysfunction and inflammation in genetically obese mice, improving overall the cardiovascular function [75,76]. Interestingly and as stated by Lefranc et al., adipocytes are poorly expressing 11b-HSD2 enzymes, responsible for the local deactivation of cortisol (in Human) or corticosterone (in rodent), suggesting that most of the detrimental effects induced by the MR activation in adipose tissues are most likely due to the binding of cortisol/corticosterone to this receptor [76]. In addition to the MR, high concentrations of GC are known to bind to the GR. GR involvement in the pathophysiology of the metabolic syndrome has been studied by using GR antagonists (RU486) that improve insulin resistance and glucose tolerance, but also via constitutive or inducible mouse models of adipose-specific *GR* KO [72,83,84,85,86,87]. Particularly, our laboratory demonstrated that the inducible adipose-*GR*-KO mice were protected against corticosterone-induced glucose intolerance and insulin resistance [72]. Moreover, the increased insulin sensitivity in adipose tissue of Adipose-*GR*-KO mice treated with corticosterone, resulted in an expansion of the fat mass, protecting individuals against liver steatosis as observed in hypercorticism conditions, such as Cushing’s syndrome [72]. Few studies document the mechanistic specificities of long-term GC impact on visceral fat. In humans, DEX pretreatment induces an increase in *GLUT4* expression and a two-fold higher rate of glucose uptake in omental adipocytes vs. subcutaneous adipocytes [88]. Interestingly, in visceral fat depot, FOXA3 binds to GR and facilitates the recognition of the GR target genes. Additionally, *FOXA3* deletion prevents visceral fat accumulation in mice subjected to a long-term DEX treatment. However, liver and muscles were not protected against GC-induced insulin resistance, uncoupling the impact of GCs on visceral fat from the overall metabolic disorder [89]. Lindroos et al. were able to demonstrate that GC-activated GR primes human adipocyte stem cells (ASC) towards adipogenesis through the expression of LIM domain only protein 3 (*LMO3)*, a transcriptional coregulator. Moreover, *LMO3* expression is increased in the VAT but not in the SCAT of obese patients and correlates with *HSD11B1* expression. Interestingly, this involvement of LMO3 in GC-mediated adipogenesis is not found in mouse preadipocytes or fat tissues, indicating species specificities [90]. Overall, in vivo studies highlighted the intricated link between insulin resistance, metabolic, and cardio-metabolic disorders induced by hypercorticism or obesity, and acting through the GC signalization [34].

### 6.2. Glucocorticoids and Insulin Resistance in Adipose Tissues at the Cellular Level

At the physiological level, under a stress or upon fasting, GC increase lipolysis in white adipocytes to mobilize energy in the form of glycerol and FFA. However, under pathological conditions, a long-term GC excess drives lipodystrophy and fat depots’ insulin resistance (Box 3). As such, 3T3-L1 cells treatment with DEX reduced the insulin-induced glucose uptake via a decreased expression of the GLUT1 transporter and a decreased translocation of GLUT4 to the plasma membrane. Additionally, DEX decreased the insulin receptor substrate (IRS-1) phosphorylation and protein expression, indicating a direct impact of GC on insulin signaling [91] (Figure 2). Additionally, acute GC treatment induces upon GR binding, the early expression of the pro-adipogenic transcription factor *C/EBP*. Second, upon priming of the cells with insulin or with the PKC inhibitor staurosporine, murine 3T3-L1 and 3T3-F442A cell lines treated with GC show an increase in the expression of *PPARG2* and of the phosphorylation of C/EBPα, two key transcription factors involved in the differentiation of white preadipocytes [92,93,94]. In contrast, in mature adipocytes, DEX treatment decreases the expression of *PPARG2*, *SREBP1C*, *C/EBPα*, and *FABP4,* indicating a decrease in anabolic pathways while increasing lipolysis (increased expression of lipases *LIPE*, *ATGL*, *MGLL*) and decreasing glucose uptake [92] (Figure 2). More recently, a study demonstrated that GC can either induce or inhibit the adipogenic differentiation program of muscle-resident fibro-adipogenic progenitors depending on the cellular concentration in cAMP [95]. Moreover, the inhibitory effects of GC-bound GR on adipogenesis were mediated through the up-regulation of a known GR target: *GILZ*. Indeed, under antiadipogenic conditions, GILZ is known to bind *PPARG2* and inhibit its expression in mesenchymal stem cells (MSCs) [95]. Overall, GCs increase but are not necessary to the recruitment of progenitor cells towards adipogenesis, while decreasing insulin sensitivity in mature cells, mirroring what can be observed in vivo [70,92]. These results were confirmed in vivo since the absence of GR in the adipose tissue of mice does not prevent normal white and brown fat development [93,96]. Interestingly, both MR and GR are known to have target genes involved in adipogenesis, lipogenesis and lipolysis. Aldosterone induces adipogenesis, while MR antagonists reduce preadipocyte differentiation in vitro [97,98]. In turn, MR promotes adipogenesis through activation of mTOR/S6K1 driving the activation of PPARγ and C/EBP signaling, but also through the activation of mTORC2 and AKT signaling [99]. GRE are found in the promoter regions of *LIPE*, *ATGL*, *MGLL*, coding for enzymes of the lipolytic pathway (Figure 2). Moreover, some of the GR-mediated effects of GC on lipolysis are thought to be mediated through ANGPTL4 and PI3KR1, leading to elevated cAMP and perilipin levels in adipocytes [100]. GCs are known to inhibit insulin-mediated glucose uptake in adipocytes, thereby defining the insulin resistance of adipose tissue [101] (Figure 2). One of the proposed mechanisms inhibiting adipocyte glucose uptake is the accumulation of reactive oxygen species (ROS) in adipocytes upon acute GC treatment. Indeed, antioxidant treatment reverses GCs’ deleterious effects in the 3T3-L1 cell line and adipose tissue insulin resistance in an obese mouse model [101]. Additionally, transcriptomic and CHIPseq analysis identified the vitamin D receptor as a direct target of GR, directly involved in GC inhibition of glucose uptake by cells [102].

### 6.3. Glucocorticoid Impact on Non-White Adipose Tissues

White VAT or SCAT are not the only fat depots responding to GCs. Chronic GC treatment decreases UCP-1 activity in the BAT in rodents and in Human, whereas an acute treatment increases brown fat activity in humans, but not in rodents [103]. Recently, BAT-specific *GR*-KO mice show no change in insulin resistance in response to diet-induced obesity, nor are losing the capacity to thermoregulate [104]. Additionally, IR KO in brown adipocytes results in an insulin secretion defect and a glucose intolerance but without insulin resistance, indicating a different metabolic impact of the insulin resistance in the BAT vs. WAT [67]. Recently, interest has been given to the characterization of the bone marrow adipose tissue (BMAT). GCs are inducing BMAT expansion. Moreover *HSD11B1*-/- mice fail to develop BMAT [105]. Interestingly, BMAT glucose uptake is non-responsive to insulin nor GC variations. However, as stated by the authors, the basal glucose uptake is higher in BMAT than it is in WAT, suggesting a non-negligible involvement of BMAT in glucose homeostasis [62].

### 6.4. Glucocorticoid Impact on the Endocrine Function Of Adipose Tissues

In clinical practice, GC are used for their anti-inflammatory and immune-suppressive properties. Adipose tissues are endocrine organs that secrete pro- or anti-inflammatory adipokines as well as adipokines involved in metabolic regulations. A high level of GC impacts the secretion of leptin and adiponectin by the adipose tissue [106]. As a feedback loop, leptin and adiponectin modulate the GC production by the HPA [107]. Knock-down of *GR* but not *MR* in human adipocytes decreases the adiponectin and leptin secretion and increases IL-6 production [108]. MR activation promotes adipose tissue inflammation and both pro-inflammatory M1 and anti-inflammatory M2 macrophage recruitment via the secretion of TNF-*α*, MCP1, and Il-6 [79,99]. In the context of insulin resistance, adipokines can have either positive or negative effects on the other tissues involved in the establishment of insulin resistance. As such, adiponectin, visfatin, and IGF1 can have a positive impact on pancreatic beta cell proliferation or function. At the opposite, leptin, apelin, TNF-*α*, and resistin can have detrimental effects (reviewed in [109]).

Box 3Summary of the Impact of GCs on Adipose Tissue.In humans and rodents, chronic GC exposure leads to adipose tissue insulin resistance, macrophage recruitment in the adipose tissues, an increase in the VAT, a reduction in the SCAT and an increase in lipolysis, characterized by FFA release in the circulation and ectopic storage of fat depots in liver, skeletal muscles and pancreas. The detrimental effects of GCs are mediated by both the GR and the MR as demonstrated by the use of pharmacological antagonists or adipocyte-specific *GR*- and *MR*-KO in vivo and in vitro. At the cellular level, GCs increase adipocyte lipolysis via an increase in the expression of lipases (LIPE, ATGL, MGLL), a decrease in the glucose uptake via the down-regulation of the GLUT1 and GLUT4 transporters, and induce insulin resistance via a decrease in IRS1 expression and activity. On the contrary, in preadipocytes, GCs have pro-adipogenic effects through a GR-mediated increase in the expression of *C/EBPβ* and *PPARG2* transcription factors (Figure 2).

## 7. Effects of Glucocorticoids on Liver Function

In the liver, insulin stimulates glucose utilization and storage through glycolysis, glycogen synthesis, and lipogenesis. Simultaneously, insulin decreases hepatic glucose production by inhibiting gluconeogenesis and glycogenolysis (Box 4).

In the liver, glucose is transported through the glucose transporter GLUT2, which is not sensitive to insulin. Glucose and insulin act synergistically in hepatic cells. Glucose can then exert some direct effects, as the translocation from the nucleus to the cytoplasm of glucokinase (GK), which phosphorylates glucose into glucose-6-phosphate. The role of insulin is to stimulate the expression and activity of enzymes implicated in glucose metabolism and to repress the activity of neoglucogenic enzymes, such as G6Pase or PEPCK, the neoglucogenesis rate limiting enzymes. Under normal conditions, after insulin treatment, AKT is activated by Ser473phosphorylation, which in turn phosphorylates and inhibits FOXO1 on Ser256, thereby suppressing *PEPCK* expression (for review, [110]).

In the liver, insulin also promotes lipogenesis, by stimulating the expression of proteins implicated in the *de novo* lipogenesis like SREBP-1c, FAS or ACC. However, these transcriptional effects require time and can be detected only 8 h after the insulin treatment [111]. Therefore, insulin favors lipogenesis mainly by acting on the enzyme activity, through phosphorylation/dephosphorylation mechanisms.

Mechanistically, the expression and activity of neoglucogenic enzymes (PEPCK and G6Pase) is controlled by FOXO1, even in a hyperglycemic context, as the activity of the transcription factor is not inhibited by AKT. The importance of FOXO1 in hepatic insulin resistance was demonstrated using transgenic animals. In fact, the invalidation of the IR specifically in the liver (LIRKO mice) resulted in hepatic insulin resistance. The concomitant invalidation of *FOXO1* was sufficient to suppress this insulin resistance [112].

As a consequence of adipose tissue insulin resistance, the lipid storage activity of the liver is maintained. More specifically, lipids originating from the adipose tissue are re-esterified in the liver. Moreover, there is an activation, which is independent of the insulin action, of the key transcription factors of the de novo lipogenesis pathway, SREBP-1c and ChREBP. This activation could result from the differential activation of AKT. In fact, the substrates of AKT are different according to its phosphorylation on Ser473 or on Thr308 residues, leading to the activation of different signaling pathways [113,114]. This is the first step of ectopic lipid accumulation in the liver, before liver steatosis, which is observed in non-alcoholic fatty liver disease (NAFLD) patients [115].

How do GCs induce insulin resistance in the liver? GCs were originally named for their ability to promote gluconeogenesis in the liver. They play central roles in the regulation of energy metabolism. In humans and in several experimental models, GCs have been shown to antagonize insulin action by inhibiting insulin-mediated glucose uptake and utilization by insulin-sensitive tissues [116,117,118]. GCs promote liver neoglucogenesis and hepatic glucose production. GCs are traditionally regarded as ‘flight and fight’ hormones, and in the fasted state they have a fundamental role to mobilize fuel for ATP generation, gluconeogenesis, glycogenolysis, and lipolysis.

Regarding glucose metabolism in the liver, GCs promote *PEPCK* gene expression, enhance its translation, and favor PEPCK enzyme synthesis and activity [119] (Figure 3). GCs stimulate *PEPCK* gene expression rapidly, between 30 and 60 min, through the induction of cAMP production, allowing the liver cells to adapt quickly to metabolic changes, through a switch from glycolysis to neoglucogenesis. However, if GCs promote *PEPCK* gene expression, the stability of *PEPCK* mRNA is decreased by GCs. This enhancement of both mRNA generation and degradation enables the liver to respond rapidly to different metabolic conditions and switch easily from catabolic to anabolic states and vice versa.

However, long-term exposure to GC results in deleterious side effects such as hyperglycemia, hepatosteatosis, and insulin resistance. This was observed in patients suffering from Cushing’s syndrome, from chronic stress, or treated chronically by GCs. The mechanisms allowing GCs to switch from glucose storage to hepatic glucose production in an acute stressful situation are the same than those promoting hepatic insulin resistance, by counteracting the effects of insulin on the liver.

These effects of GCs have been extensively described in different in vivo models, mainly in rodents in which the concentration of circulating GCs was induced or reduced, either by modulating 11b-HSD1 activity, by grafting pellets of GCs or by directly injecting DEX in mice or rats. Other studies have chosen to directly act on the GR, in order to modulate the GC signaling pathway and to analyze the consequences on metabolism and hepatic function.

*HSD11B1* is highly expressed in the adipose tissue, the brain, but also in the liver. It has been observed that the level of expression and activity of the enzyme is increased in T2D patients, as well as in rodent models of diabetes (hyperphagic diabetic *db/db* mice and diabetic Goto–Kakizaki rats) [120,121,122]. To get insight in the role of 11b-HSD1 in the development of insulin resistance and T2D diabetes, different strategies of gain or loss of function were developed.

The total invalidation of *HSD11B1* or its pharmacological inhibition in rodent has demonstrated an improvement in glucose tolerance, insulin sensitivity as well as reduced body weight gain [123,124]. If mice with a specific liver invalidation of *HSD11B1* presented no metabolic effect, the specific hepatic overexpression of the enzyme led to normal glucose tolerance in mice with hyperinsulinemia and dyslipidemia and an increase in hepatic fat content. The analysis of gene expression profile showed an increased expression of key lipogenic genes such as *FAS*. If *SREBP-1c* expression was not altered, the expression of Liver X receptors (*LXR*), which controls *FAS* expression, was induced [125,126].

When DEX was injected in rats (1 mg/kg/day administration for five days), as expected, the animals became insulin-resistant. Regarding insulin signaling pathway in the liver, the protein level of IR was unchanged, whereas the levels of IRS1 and PI3K protein were increased [127]. In spite of this increased expression, protein phosphorylation was dramatically decreased by 40% for IR and 60% for IRS1, and the PI3K activity was decreased by 80%, thus dramatically inhibiting insulin signal [127].

This inhibition of insulin signaling prevents AKT activation and then FOXO1 inhibition, which can then stimulate the expression of its neoglucogenic target genes.

When GCs bind to GR, the complex translocates from the cytoplasm to the nucleus, where it binds on the DNA to GRE or nGRE. It seems that the anti-inflammatory effects of GCs are mainly linked to the repression of inflammatory genes, whereas the metabolic effects of GCs are more likely linked to the activation of gene expression (e.g., *PEPCK* gene) [128].

Liver-specific deletion or pharmacological blockade of the GR has been shown to have beneficial effects, including decreased fasting plasma glucose and insulin levels and reducing the progression to diabetes. Furthermore, anti-sense oligonucleotides directed against the *GR* have shown liver-specific effects, decreasing fasting hyperglycemia, reducing fasting insulin levels and decreasing GC-stimulated hepatic glucose output [129,130,131,132].

Concerning *PEPCK* gene, two GREs have been identified on its promoter [133]. Different GR-associated binding proteins and related receptors have also been implicated in modifying GR action. To fully induce gene expression, GR acts in synergy with different factors, such as hepatocyte nuclear factor-*α*, cAMP response element-binding protein, C/EBP*β*, transcriptional coactivator 2 (TORC2), FOXO1, peroxisome proliferator-activated receptor γ-coactivator 1-*α* (PGC1*α*), mediator subunit-1 (MED1), and steroid receptor coregulator-1 (SRC-1) [134]. *PGC1α* expression is repressed via the AKT pathway in response to insulin [135,136], and under GR activation, it participates to the onset of hyperglycemia, through inducing expression of hepatic neoglucogenic genes. Indeed, PGC1*α* expression is necessary and sufficient to promote expression of *PEPCK* gene both in vivo and in vitro in cultured primary hepatocytes, but the addition of DEX alone or in combination with cAMP strongly potentiates its effects [137]. However, the role of PGC1*α* on hepatic metabolism was recently revisited, as Besse-Patin et al. demonstrated that PGC1*α* regulates the expression of *IRS1/2* in hepatocytes, preparing the response to insulin during the fast-to-fed transition [138].

MED1 is a receptor binding protein important for normal GR function in the liver. Liver-specific deletion of *MED1* protects against DEX-induced hepatic steatosis, which is a hallmark of hepatic insulin resistance [139].

Recently, a new partner of the GR, the protein E47, has been identified as a GR modulator and has been implicated in hepatic lipid and glucose metabolism. Indeed, *E47*-invalidated mice are protected from steroid-induced hyperglycemia, dyslipidemia and hepatic steatosis. GR and E47 were found to bind to the promoters of metabolic genes like *glycerol-3-phosphate acyltransferase, GK, PEPCK, G6Pase* and *insulin-like growth factor binding protein-1* [140].

GC treatment can also act on hepatic metabolism through the activation of other targets. Indeed, in a rat model of Cushing’s syndrome, animals were adrenalectomized bilaterally and pellets containing 100 mg of corticosterone were implanted under the skin. In these animals, insulin concentrations were higher, with no difference in blood glucose, indicating that these animals were insulin-resistant. They exhibited high plasma levels of leptin, cholesterol, and TAG. In the liver, the lipid content was increased, and the authors could observe an increase in adenosine monophosphate-dependent kinase (AMPK) activity. This in vivo hepatic activation of AMPK was a direct effect of GC, as it could be reproduced in HepG2 cells [141]. AMPK is an energy sensor [142] and is activated by a decrease in ATP and a concomitant increase in AMP cellular content. Once activated, AMPK switches off anabolic pathways such as fatty acid, TAG and cholesterol synthesis, as well as protein synthesis and transcription, and switches on catabolic pathways, including glycolysis and fatty acid oxidation.

The AMPK pathway is involved in GC-induced regulation of fatty acid utilization. In fact, DEX treatment could activate AMPK and improve hepatic lipogenesis, whereas the inhibition of AMPK by compound C suppresses hepatic lipogenesis induced by DEX [143]. Then, AMPK, as a target of GR, transduces the lipogenic/pro-steatosic effect of GC observed in insulin-resistant liver.

Liver X receptors (LXRs) are members of the nuclear hormone receptor superfamily and have crucial roles in cholesterol and lipid metabolism. They are transcription factors involved in the de novo lipogenesis through the induction of *SREBP1c* expression. There are two isoforms (LXRα and β) and *LXRβ*-KO mice are protected from hepatic steatosis despite elevated circulating GC levels and this has raised the potential of LXRβ as a therapeutic target to treat and prevent GC-induced hepatic steatosis [144]. This hypothesis was tested through the administration of LXRβ specific antagonist (GSK2033) to mice. The authors could observe that the blockade of LXRβ could also inhibit the hepatic insulin resistance induced by DEX administration in mice. The antagonist blocked the GC-GR pathway at different levels, by decreasing GR translocation into the nucleus and the recruitment of GR co-activators on the *PEPCK* promoter. The full mechanisms are not completely understood and still have to be deciphered [134].

Functional regulations of GCs on mitochondria have also been reported. Mitochondria are the center of energy supply in cells, providing more than 95% of ATP required for cell metabolism. GCs, via their action on hepatocyte mitochondria, are implicated in the hepatocyte cellular defect, which can initiate or amplify hepatic insulin resistance. This deleterious action of GCs on hepatic mitochondria was recently observed in the liver of DEX-treated mice [145]. The authors reported an alteration in mitochondrial DNA (mtDNA), which led to a mitochondrial dysfunction, with an increased ROS production and a decreased ATP production, causing a cellular dysfunction. In vitro, DEX treatment on isolated hepatocytes induced an increase in ROS levels, reduced ATP synthesis, opened the mitochondrial transition pore and damaged mtDNA. Moreover, DEX treatment causes the disruption of the structure and function of hepatocyte mitochondria [145].

Recently, it was observed that in response to hepatic insulin resistance (LIRKO mice), hepatic cells secrete a serine proteinase inhibitor, SERPINB1, which stimulates pancreatic beta cell proliferation in mouse and human islets. The purpose of the study was to understand the mechanisms allowing pancreatic islet hyperplasia in this mouse model [146]. Since then, an increased plasma level of SERPINB1 has been described in type 2 diabetic patients [147]. Until now, no relation has been made between SERPINB1 and GC. However, since in our mouse model of severe insulin resistance induced by GC, an increase in pancreatic beta cell proliferation was observed [49], it could be interesting to study whether GC can promote hepatic secretion of SERPINB1.

Box 4GC Effects on Liver Function.The physiological neoglucogenic role of GC is essential for the transition from an anabolic to a catabolic state, during fasting, but when their production is deregulated, or when the GR pathway is over-activated, the physiological actions are disrupted, leading to liver insulin resistance with glucose overproduction and increased blood glucose levels with an associated lipogenesis, causing hepatosteatosis. In the liver, the activation of the GC-GR signaling pathway inhibits the IR pathway and Akt activity and induces FOXO1, which in turn stimulates *PEPCK* and *G6Pase* expression and, ultimately, glucose production (Figure 3).

## 8. Effect of Glucocorticoids on Skeletal Muscles

Skeletal muscles account for 40% of whole-body mass in a lean subject and are responsible for most of insulin-stimulated glucose uptake [148]. Skeletal muscles are also major storage sites of amino acids that are substrates for hepatic glucose production through neoglucogenesis. Thus, skeletal muscles are highly involved in glucose homeostasis and their resistance to insulin represents a key pathogenic event in the alteration of total glucose uptake in insulin resistant as well as in diabetic patients. In parallel, skeletal muscles are also targets of GCs, as illustrated by the dramatic muscle atrophy or wasting during chronic GC exposure, such as Cushing’s syndrome [149]. More precisely, GCs are known as potent antagonists of insulin action on skeletal muscles and the underlying molecular mechanisms have been investigated during the past decades. Globally, as GCs are catabolic hormones required to mobilize storage to provide energy for the organism (to respond to danger, stress or metabolic imbalance), their action on muscle is to decrease glycogen stores and degrade proteins into amino acids, substrates for hepatic glucose production (Box 5 and Figure 4).

Muscles are major sites for glucose uptake, utilization, and storage (in the form of glycogen). Glucose is transported from the blood to the cytosol of muscle cells through the specific glucose transporter GLUT4. Upon insulin stimulation, GLUT4 enriched vesicles are very quickly translocated from the cytosol to the membrane, leading to increased GLUT4 density and glucose transport. Studies have previously demonstrated that GC decrease glucose uptake in skeletal muscles through the inhibition of GLUT4 translocation induced by insulin [150]. GC also reduces glycogen synthesis in muscle cells through suppression of the activity of glycogen synthase, the enzyme that catalyzes glycogen storage from glucose [151]. GC act on insulin signaling through the binding to their receptors as shown by studies revealing a strong correlation between *GR* mRNA levels in the muscle and insulin resistance in diabetic patients [152]. Moreover, treatment of diabetic patients to normalize insulin sensitivity also results in the normalization of *GR* expression in muscles, suggesting a strong link in between GR signaling and insulin sensitivity [122,152].

Additionally, GCs have been shown to reduce PI3Kinase activity in myotubes in vitro [127]. Rats treated with DEX present reduced IR tyrosine phosphorylation in skeletal muscles [153], and decreased AKT phosphorylation in rat muscles and in a cell line of myotubes, the C2C12 cells [154]. Otherwise, GR is able to compete with IRS1 for its association with PI3kinase subunits p11 and p85, leading to reduced PI3Kinase activity and level of AKT phosphorylation [155]. Apart from direct interaction of GR with molecules involved in insulin signaling, deciphering the action of GC on muscle cells requires to identify genes that are activated or inhibited by GC treatment, since GC/GR complexes are potent transcription factors. Such study was performed and led to the identification of 173 potent targets of GC/GR in C2C12 cells, with some of the identified candidates involved in the insulin signaling pathway, such as *p85α-PI3K* [156].

Concerning glycogen storage in muscles, a specific kinase called glycogen synthase kinase 3 (GSK3) phosphorylates and inhibits glycogen synthase. When insulin is elevated, AKT is activated and phosphorylates GSK3 to reduce its activity, leading to glycogen formation [151]. In contrast, GC are able to decrease the level of phosphorylated GSK3 resulting in an increased level of its dephosphorylated and active form, thus inhibiting glycogen synthase. Alternatively, GC play a permissive role for catecholamine action on muscles. GC are indeed required for the reduction of glycogen synthase activity induced by catecholamine treatment in rats [157].

In a mouse model of hyperphagic and insulin resistant mice, *ob/ob* mice, GC blood levels are elevated [158]. Normalization of such levels by adrenalectomy in those mice restores muscle insulin sensitivity. In mice fed with a high fat diet, muscle insulin sensitivity is impaired and the use of a GR antagonist, RU38486, is sufficient to rescue muscle insulin sensitivity. Finally, in the muscle, the level of the receptor for the insulin-sensitizing adipokine adiponectin is reduced upon GC treatment [159].

Due to this interconnected control of insulin signaling by GCs, studies have been conducted to define whether GC inhibition may improve insulin sensitivity, especially in skeletal muscles. As the enzyme 11b-HSD1 controls the levels of active GC, 11b-HSD1 inhibitors have been used in mice to define whether insulin sensitivity was increased. Actually, treatment of mice with a specific 11b-HSD1 inhibitor led to decreased blood glucose levels through decreased levels of the inhibitory form of IRS1 (phosphorylated on Serine 307) and increased levels of the activated form of AKT [160].

Among molecules that are controlled by GC and may be involved in the control of insulin signaling, myostatin has drawn a particular interest. Myostatin is a negative regulator of muscle growth. It controls muscle cell differentiation [161]. Invalidation of *myostatin* leads to mice with hypertrophied muscles [162]. Interestingly, myostatin can inhibit AKT activation [163], and GC stimulate the expression of myostatin, suggesting a role for this myokine in the GC-induced inhibition of AKT (Figure 4). It has been proposed that GC directly induce *myostatin* expression since they can increase the activity of *myostatin* promoter [164]. An indirect control of myostatin by GCs has also been suggested and may involve C/EBP transcription factors [165]. Interestingly, the deletion of *myostatin* gene strongly reduced the muscle atrophy induced by GC chronic treatment. Thus, myostatin is clearly implicated in the impact of GCs on muscle atrophy, a role that may implicate a reduction of PI3K/AKT signaling pathway [166]. However, myostatin has not yet been clearly involved in insulin sensitivity.

Box 5GC Effects on Skeletal Muscles.In muscles, GCs decrease insulin action through many molecular targets such as IRS-1, PI3kinase, AKT, GSK3. Altogether, these proteins are modified by GC exposure ultimately leading to decreased GLUT4 translocation to the plasma membrane, decreased glucose transport, and protein catabolism (Figure 4).

## 9. Glucocorticoid-Induced Insulin Resistance in Other Tissues

### 9.1. Bone

A prolonged exposure to GCs, whether its related to a Cushing’s syndrome or to GC treatment, induces insulin resistance in bone cells, translating into an increase in bone marrow adiposity and osteoporosis [167]. In turn, GCs control the differentiation and function of osteoblasts, osteocytes, the bone forming cells and of osteoclasts, the bone resorbing cells [167]. At high doses, GCs impair the expression of genes involved in the WNT signaling pathway, inhibiting the differentiation of precursor cells into osteoblasts. Additionally, GCs induce the production of the receptor activator of nuclear factor kappa-B ligand (RANKL) increasing osteoclast formation, activation, and survival, while inhibiting the production of osteoprotegerin (OPG), an inhibitor of RANK/RANKL signaling. GC also stimulate osteoblast and osteocyte apoptosis through activation of proapoptotic factors of the BCL-2 family [167]. Interestingly, insulin resistance induced by the deletion of one allele of the *IR* in osteoblasts decreased insulin sensitivity of the muscle and the WAT, highlighting the complex crosstalk that exists between insulin-sensitive tissues. Moreover, GCs induce a decrease in the circulating levels of osteocalcin, a hormone participating in the insulin sensitivity of different tissues [168]. Recently, circulating osteocalcin levels have also been shown to be dysregulated in humans rendered insulin-resistant after GC treatment [169].

### 9.2. Gut

In response to insulin, enterocytes internalize GLUT2 from the basolateral site of enterocytes in order to prevent further transport of glucose in the blood during hyperglycemia. However, in insulin-resistant animals, GLUT2 trafficking is disturbed and the enterocytes lose the capacity to limit glucose absorption [170,171]. Moreover, obesity induces systemic- but also gut-specific inflammation, via the release of pro-inflammatory cytokines by T-cells, which contribute to enterocyte insulin resistance in humans and rodents [172,173]. Indeed, the gut immune system regulates glucose metabolism, notably through the impacted release of gut-derived hormones, such as Glucagon-Like Peptide-1 (GLP-1), known to promote insulin secretion by pancreatic beta cells. As such, TNF-reduces GLP-1 secretion by enteroendocrine cells [174]. Moreover, local inflammation induced by obesity participates to modifications of the gut microbiota and to altered epithelial barrier, worsening systemic inflammation and metabolic disorders [174]. In this context, it is therefore not surprising that synthetic GC treatment or activation of the HPA axis impact positively on enterocyte function and local inflammation [175]. Interestingly, an extra adrenal production of GCs is found in several tissues including immune system, adipose tissue, brain and intestine [176]. The gut microbiota also plays a role in the diurnal synthesis of corticosterone by intestinal epithelial cells (IECs). Indeed, interactions between bacteria and Toll-like receptors located on IECs decrease the expression of *PPAR* that results in the down-regulation of corticosterone throughout the day. Accordingly, mice kept in a germ-free environment or chronically treated with antibiotics develop hyperglycemia and insulin resistance because of the sustained production of corticosterone in the gut [177].

In addition to its effect on insulin sensitivity in gut, GC have also a specific effect on the incretin GLP-1. Indeed, GLP-1 is released by enteroendocrine cells and improves the glucose-induced insulin secretion by pancreatic beta cells. Both in vivo and in vitro, GC decrease the secretion of GLP-1 in response to glucose by enteroendocrine cells [178].

### 9.3. Brain

The brain is an insulin-sensitive organ. IR are expressed in all brain cell types, from neurons to endothelial cells, but the density and subunits of the receptor (alpha or beta) vary between regions, with the highest expression found in the cerebellum, hippocampus, cerebral cortex, striatum, olfactory bulb, and hypothalamus. Despite this wide representation of IR among the brain, insulin acts differently on the brain than on peripheral organs. For example, the brain represents only 2% of body weight in humans but consumes about 20% of glucose-derived energy (neurons are the cellular type with the highest energy demand) [179]. Studies show that GLUT4, an insulin-dependent glucose transporter, is also co-expressed in some brain regions relative to learning, such as the hippocampus [180] or the hypothalamus [180], suggesting the importance of an insulin-stimulated glucose influx in these regions for metabolic control. In fact, mice lacking *GLUT4* in the central nervous system have a reduced glucose uptake in the brain, but they are also glucose intolerant, present a hepatic insulin resistance and impaired glucose sensing, demonstrating the importance of *GLUT4* expression in the brain for peripheral glucose sensing and glucose metabolism [181]. Nevertheless, the main glucose uptake in the brain is insulin-independent, through the glucose transporter GLUT3 [182], demonstrating another role for insulin in the brain apart from its hypoglycemic effect. In the brain, insulin promotes neurite outgrowth, synaptic plasticity, modulates catecholamine trafficking, the efficacy of synaptic transmission (long-term enhancement or reduction), and inhibits neuron apoptosis. Through its action on hypothalamus, insulin has systemic metabolic effects by suppressing hepatic glucose production [183], lipolysis in adipose tissue [184], and by decreasing food intake [185,186,187].

Of interest is the origin of insulin in the brain: if the majority of studies designates pancreatic beta cells as the source of brain’s insulin, a few and controversial works suggest a de novo synthesis in the brain [188,189]. Pancreatic insulin is believed to penetrate brain thanks to a specific, saturable carrier located on capillary endothelial cells [190]. An interesting study using human adipose tissue-derived microvascular endothelial cells shows that uptake of a fluorescent-tagged insulin was saturable and competed by S961, an IR antagonist, suggesting that the microvascular brain insulin carrier could be the IR [191]. A princeps study investigated the effects of DEX on insulin transport into the brain in vivo and showed that DEX decreased insulin uptake by 49%. In link with their previous finding that the Km (Michaelis-Menten constant which measures the kinetics of an enzyme reaction) of the still unknown insulin transporter is comparable to the Kd (dissociation constant which reflects affinity of a ligand for its receptor) of IR, they hypothesized that insulin transport into the brain is mediated via an IR-related mechanism that is sensitive to inhibition by GC [192]. Since insulin action on hypothalamus decreases food intake and helps maintaining weight, GC-induced deficiency in insulin transport to the brain may mediate some of the metabolic effects of GC excess, like weight gain, adipose tissue expansion, and increased production of hepatic glucose.

Chronic exposure to GC is known to impair neural plasticity, in particular synaptic plasticity which is at the basis of learning and memory capacities [193,194,195]. In addition, growing evidences linked a decrease in cognitive function and an increase of neuropsychiatric disorders, such as Alzheimer’s disease, to T2D and brain insulin resistance (defined as a decrease in brain IR activity and sensitivity) [182]. Piroli et al. demonstrated that short-term (one-week) corticosterone administration to rats does not reduce *IR* expression in the hippocampus but significantly reduces IR activity, with reduced insulin-stimulated phosphorylation of the IR and decreased total AKT and total GLUT4 protein expression. They propose that the GC-induced insulin resistance in the brain could lead to a deficit in hippocampal plasticity, allowing hippocampal dysfunction observed in patients suffering from Cushing’s syndrome, T2D, or Alzheimer disease [196,197]. Likewise, Stranahan et al. showed in two models of diabetic rodents, insulin-deficient (streptozotocin-treated) rat and insulin-resistant (*ob/ob*) mice, a reduced hippocampal synaptic plasticity and adult neurogenesis, leading to impaired hippocampus-dependent memory. As these two models present elevated GC levels, they elegantly demonstrated that defects in hippocampal synaptic plasticity are reversed under restoration of normal GC levels. Of interest is that administration of high doses of corticosterone to adrenalectomized mice reinstates learning deficit, suggesting that corticosterone is the major contributor to hippocampal impairment and no other adrenal factors. Stranahan et al. then hypothesized that cognitive impairment in diabetes may result from a GC-mediated deficit in neurogenesis and synaptic plasticity [198]. In humans, the same pattern of association between insulin resistance, elevated GC levels and alteration of cognitive functions, such as memory, is observed [199,200,201]. For example, patients with Cushing’s syndrome display memory, cognition and attention impairment [202]. However, a cohort study upon middle-aged T2D patients reported that only subjects with diabetic complications present an enhanced cortisol secretion [203]. To conclude, a strong correlation between GC and a decline in cognitive functions has been observed; however, the precise mechanisms are still not deciphered, and it is still not clear whether these effects of GC are linked to GC-induced brain insulin resistance.

## 10. Conclusions

GCs are potent inducers of insulin resistance in all insulin-sensitive tissues. They are catabolic hormones and modify the insulin pathway that is the main anabolic pathway. Molecular mechanisms to do so are multiple and imply several molecules from the insulin receptor to transcription factors. Yet, the imbalance in the insulin sensitivity induced by GC may be compensated if beta cells are able to adapt and produce more insulin (Figure 5). Further research should then focus on how new molecules with therapeutic efficiency similar than GC but without side effects could be developed. In parallel, the mechanisms underlying beta cell adaptation to GC-induced insulin resistance require to be investigated, to prevent glucose homeostasis dysregulations triggered by GC.

## Figures and Tables

**Figure 1 ijms-22-00623-f001:**
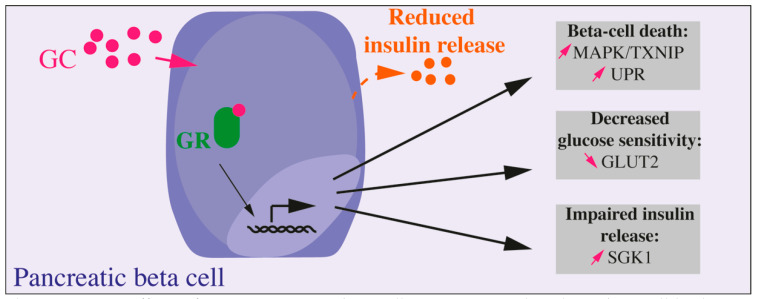
Direct effects of GC on pancreatic beta cell. GC exposure directly on beta cell leads to altered beta cell function, decreased glucose sensitivity, and insulin secretion and, *in fine*, to beta cell death. Most of the presented results were obtained in vitro, on isolated islets, primary beta cells, or beta cell lines.

**Figure 2 ijms-22-00623-f002:**
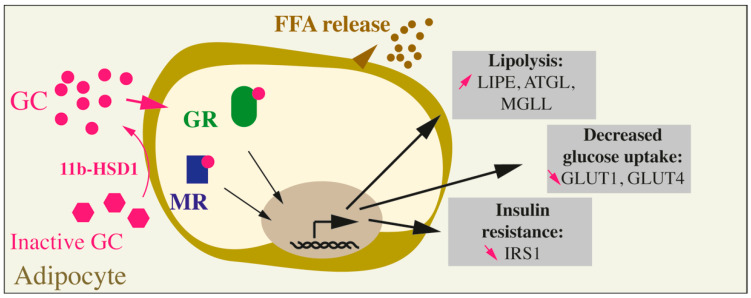
Effects of GC on adipocytes. Inactive GC are activated through the action of the specific enzyme 11b-HSD1. GCs can then bind to their receptors (GC receptor, or GR) or to the mineralocorticoid receptor (MR). The overall action of GCs on mature adipocytes leads to an increase in lipolysis (augmented LIPE, ATGL, and MGLL), a decrease in glucose uptake (decreased GLUT1 and GLUT4) and insulin resistance (reduced IRS1).

**Figure 3 ijms-22-00623-f003:**
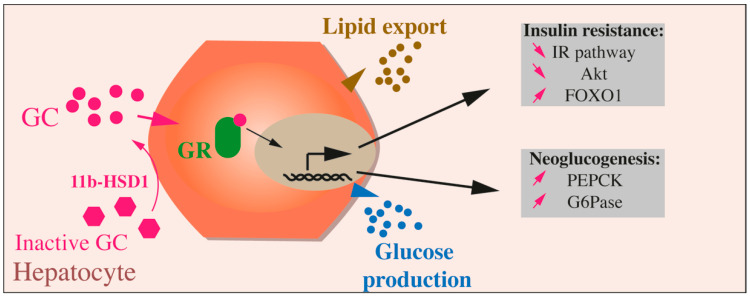
Effects of GCs on hepatocytes. Inactive GCs are activated through the action of 11b-HSD1. GCs binding to their receptor (GC receptor, or GR) leads in hepatocytes to glucose production through upregulation of neoglucogenesis (augmented PEPK and G6Pase), to lipids export and insulin resistance through reduced IR pathway, Akt activation, and increased FOXO1.

**Figure 4 ijms-22-00623-f004:**
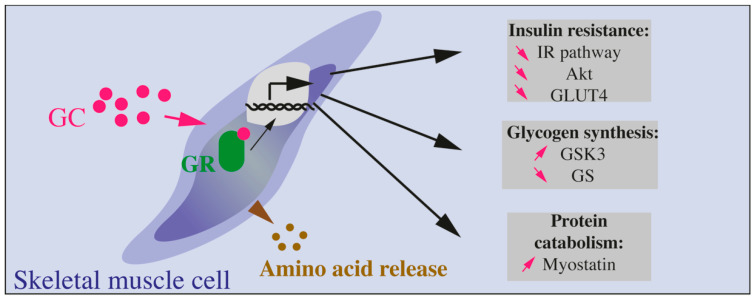
Effects of GCs on skeletal muscle cells. GCs bind to their receptor (GC receptor, or GR) and lead to protein catabolism (involving increased myostatin production) and amino acids release. Glycogen synthesis is reduced through increased GSK3 and reduced GC. Finally, insulin resistance is also induced by GCs through reduced Akt, GLUT4, and IR pathway.

**Figure 5 ijms-22-00623-f005:**
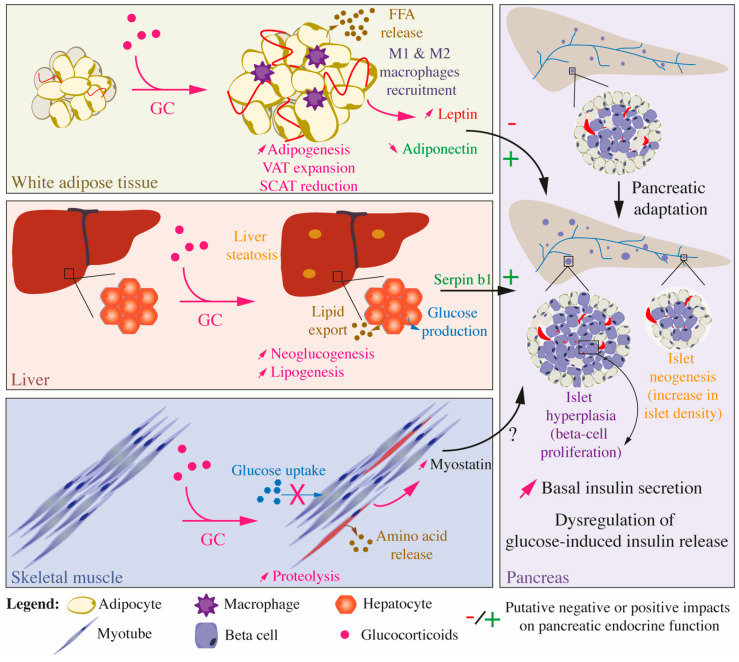
Integrated effects of glucocorticoids on glucose and lipid homeostasis and inter-organ communication. Impacts of GC treatment on the white adipose tissue, liver, and skeletal muscles, which are the main organs and tissues involved in the establishment of insulin resistance, as well as on the pancreas, which secretes insulin, are presented. Upon GC treatment, WAT, liver, and skeletal muscles develop insulin resistance but also influence each other and the pancreatic endocrine function through the secretion of peptides, adipokines, and myokines. Chronic exposure to GCs is associated with a pancreatic adaptation, characterized by an increase in basal insulin secretion, augmented beta cell proliferation leading to islet hyperplasia, and increased islet density, an indirect evidence of islet neogenesis.

## Data Availability

Not applicable.

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
