# Peer review of "Molecular Mechanisms of Glucocorticoid-Induced Insulin Resistance"

_ijms, 2021, doi:10.3390/ijms22020623_

Round 1
Reviewer 1 Report
The MS entitled “Molecular Mechanisms of glucocorticoids-induced insulin resistance” by Beaupère et al. is an extensive revision on the literature dealing with the effects of GC in different tissues leading to insulin resistance and pancreatic defects. The review is well-organized, well-written and informative. It is easy to read and the summary boxes at the end of each section are useful as take-home messages. Even though I have not been able to see the Figure, I have easily followed all the information.
Minor points:
-The paragraph at lines 312-325 is at the end of the section dialing with effects at the pancreas level but is actually an introduction to the section devoted to insulin resistance in the different tissues. It should have a title.
-Description of the functions of the adipose tissues in lines 333-334 and 353-356 is a bit repetitive and could be combined in a single listing.
Author Response
Review report (reviewer 1)
“The MS entitled “Molecular Mechanisms of glucocorticoids-induced insulin resistance” by Beaupère et al. is an extensive revision on the literature dealing with the effects of GC in different tissues leading to insulin resistance and pancreatic defects. The review is well-organized, well-written and informative. It is easy to read and the summary boxes at the end of each section are useful as take-home messages. Even though I have not been able to see the Figure, I have easily followed all the information.
Minor points:
-The paragraph at lines 312-325 is at the end of the section dealing with effects at the pancreas level but is actually an introduction to the section devoted to insulin resistance in the different tissues. It should have a title.
-Description of the functions of the adipose tissues in lines 333-334 and 353-356 is a bit repetitive and could be combined in a single listing.
Answers to reviewer 1
We thank reviewer 1 for her/his helpful comments. We agree that the paragraph at lines 312-325 required a title. Thus, we added the following title:” A BRIEF MOLECULAR VIEW ON INSULIN SIGNALLING PATHWAY”. It helps clarify the organization of the manuscript.
Concerning the 2nd point, the sentence “The four main functions of the adipose tissue are: 1) energy storage in the form of TAG, 2) TAG mobilization to release free fatty acids (FFA) and glycerol, 3) production and secretion of adipokines and 4) thermoregulation” has been replaced by “The four main functions of the adipose tissue are energy storage in the form of TAG, release of free fatty acids (FFA) and glycerol, secretion of adipokines and thermoregulation.”
Reviewer 2 Report
In this review, the authors show a variety of actions of Glucocorticoids (GC), steroids secreted by the adrenal cortex via the hypothalamic pituitary-adrenal axis control. The neuroendocrine system is involved in tissue repair, immune stability and metabolic processes, lipid and protein metabolism. In this paper, they especially described GC-induced insulin resistance in adipose tissue, liver, and muscle, identified by evidence suggested with animal model studies.
Generally, this review is well written and informative. However, the present manuscript form is difficult to understand the details of actions of GCs. For readers, illustrations displaying important points on each topic should be included. For example, receptors (GR, MR and their different contribution), long-term or chronic exposure to GCs, paradoxical effects of GCs in vitro and in vivo,… etc. Inclusion of illustration demonstrating the summary in each topic will make the paper more attractive one for readers.
Author Response
Review report (reviewer 2)
“In this review, the authors show a variety of actions of Glucocorticoids (GC), steroids secreted by the adrenal cortex via the hypothalamic pituitary-adrenal axis control. The neuroendocrine system is involved in tissue repair, immune stability and metabolic processes, lipid and protein metabolism. In this paper, they especially described GC-induced insulin resistance in adipose tissue, liver, and muscle, identified by evidence suggested with animal model studies.
Generally, this review is well written and informative. However, the present manuscript form is difficult to understand the details of actions of GCs. For readers, illustrations displaying important points on each topic should be included. For example, receptors (GR, MR and their different contribution), long-term or chronic exposure to GCs, paradoxical effects of GCs in vitro and in vivo,… etc. Inclusion of illustration demonstrating the summary in each topic will make the paper more attractive one for readers.”
Answer to reviewer 2
We thank the reviewer for her/his helpful comments. We agree that the manuscript would benefit from more specific figures. Thus, we have now added figures to illustrate the GC effects for each of the target tissues, with the molecular impacts, and we simplified the last figure to illustrate inter-organ communication. We hope that these new figures will satisfy your request.

Round 2
Reviewer 2 Report
This version is very good.